# Scalable and flexible inference framework for stochastic dynamic single-cell models

**Sebastian Persson**[1,2], **Niek Welkenhuysen**[1,2], **Sviatlana Shashkova**[3,4],
**Samuel Wiqvist**[5], **Patrick Reith**[1,2,6], **Gregor W. Schmidt**[7], **Umberto Picchini**[1,2]*,
**Marija Cvijovic**[1,2]*

**1** Department of Mathematical Sciences, University of Gothenburg, Gothenburg, Sweden, **2** Department of
Mathematical Sciences, Chalmers University of Technology, Gothenburg, Sweden, **3** Department of
Microbiology and Immunology, Institute of Biomedicine, Sahlgrenska Academy, University of Gothenburg,
Gothenburg, Sweden, **4** Department of Physics, University of Gothenburg, Gothenburg, Sweden, **5** Centre for
Mathematical Sciences, Lund University, Lund, Sweden, **6** Department of Biology and Biological Engineering,
Chalmers University of Technology, Gothenburg, Sweden, **7** Department of Biosystems Science and
Engineering, ETH Zurich, Basel, Switzerland

* picchini@chalmers.se (UP); marija.cvijovic@chalmers.se (MC)

UNITED STATES

**Data Availability Statement:** The.tif files of the the
experimental data are available on figshare https://
figshare.com/s/2d56e0a6a928ef1dd7ac. The time-
lapse data used to fit the Mig1-model are available

## Abstract

Understanding the inherited nature of how biological processes dynamically change over
time and exhibit intra- and inter-individual variability, due to the different responses to envi-
ronmental stimuli and when interacting with other processes, has been a major focus of sys-
tems biology. The rise of single-cell fluorescent microscopy has enabled the study of those
phenomena. The analysis of single-cell data with mechanistic models offers an invaluable
tool to describe dynamic cellular processes and to rationalise cell-to-cell variability within the
population. However, extracting mechanistic information from single-cell data has proven
difficult. This requires statistical methods to infer unknown model parameters from dynamic,
multi-individual data accounting for heterogeneity caused by both intrinsic (e.g. variations in
chemical reactions) and extrinsic (e.g. variability in protein concentrations) noise. Although
several inference methods exist, the availability of efficient, general and accessible methods
that facilitate modelling of single-cell data, remains lacking. Here we present a scalable and
flexible framework for Bayesian inference in state-space mixed-effects single-cell models
with stochastic dynamic. Our approach infers model parameters when intrinsic noise is mod-
elled by either exact or approximate stochastic simulators, and when extrinsic noise is mod-
elled by either time-varying, or time-constant parameters that vary between cells. We
demonstrate the relevance of our approach by studying how cell-to-cell variation in carbon
source utilisation affects heterogeneity in the budding yeast *Saccharomyces cerevisiae*
SNF1 nutrient sensing pathway. We identify hexokinase activity as a source of extrinsic
noise and deduce that sugar availability dictates cell-to-cell variability.

## Author summary

Understanding the causes of heterogeneity and the means by which it can be controlled is
crucial for manipulating cellular populations and treating diseases. To this end, single-cell

on GitHub; https://github.com/cvijoviclab/PEPSDI/
tree/main/Intermediate/Experimental_data/Data_
fructose. PEPSDI is available on GitHub (https://
github.com/cvijoviclab/PEPSDI). The code for
producing the results in this paper is further
available in the GitHub repository (https://github.
com/cvijoviclab/PEPSDI/tree/main/Paper/
Examples).

**Funding:** This work was supported by the Swedish
Research Council (VR2019-03924 to UP and
VR2017-05117 to MC), the Chalmers AI Research
Centre (CHAIR) to UP, the Swedish Foundation for
Strategic Research (FFL15-0238 to MC) and the
Marie Skłodowska-Curie grant agreement No
764591 to PR. The funders had no role in study
design, data collection and analysis, decision to
publish, or preparation of the manuscript.

**Competing interests:** The authors have declared
that no competing interests exist.

time-lapse microscopy data is often combined with dynamic modelling. However, the
construction of mechanistic models requires the ability to infer unknown model quanti-
ties from data, while simultaneously accounting for intrinsic and extrinsic noise. Here we
propose a Bayesian inference framework which enabled us to elucidate sources of cell-to-
cell variability in yeast signalling and provides deeper insights into the causes and conse-
quences of heterogeneity. Our approach is versatile and can for example further be applied
in pharmacokinetic and pharmacodynamic studies, epidemic studies, as well when
modelling the behaviour of cancer cell populations.

This is a *PLOS Computational Biology* Methods paper.

## Introduction

Traditionally, investigations in the life sciences have focused on a population "ensemble aver-
age" level. On one side, such population approach reduces noise from atypical cells. However,
any cellular population is in general heterogeneous, with a range of different physical, chemi-
cal, and biological properties. Thus, population methods smooth out and hence miss biologi-
cally relevant cell-to-cell variability [1]. For example, such approaches will overlook drug-
resistant bacteria or cancer cells in a general cell population. Furthermore, cell-to-cell variabil-
ity plays an important role in the decision making of a population, such as quick adaptation to
fluctuating environments [2]. The only way to identify all biologically relevant processes, and
thus describe cell heterogeneity, is to investigate the whole population cell-by-cell [3].

To study life processes occurring in individual cells within the population, fluorescent
time-lapse microscopy can be employed to track proteins in multiple cells over time [1]. Ide-
ally, this can give a view of cell heterogeneity, and potentially help elucidate cellular reaction
dynamics. However, to learn more from acquired data, dynamic modelling naturally comple-
ments time-lapse microscopy, and aids in deducing sources of cell-to-cell variability [4–6]. But
to fully exploit dynamic modelling, unknown model parameters must typically be inferred/
estimated from data [7]. This is non-trivial to perform from single-cell time-lapse data, mainly
because models describing individual cells must account for cell-to-cell variability caused by
both intrinsic (e.g., variations in chemical reactions) and extrinsic (e.g., variability in protein
concentrations) noise [8].

Several inference methods exist that, to various degrees, account for cellular heterogeneity
when inferring model parameters from time-lapse data. In common, they allow extrinsic noise
to be modelled by letting model parameters, e.g protein synthesis rates, vary between cells [4,
6, 9, 10]. Methods based on ordinary differential equations (ODEs) [6] further assume that
intrinsic noise is negligible. On the other side, the dynamic prior propagation (DPP) [4] and
the stochastic differential equation mixed-effects models (SDEMEM) [9, 10] encode intrinsic
noise via exact [11], or approximate [12] stochastic simulators, respectively.

Although useful, current inference methods have drawbacks. The fact that ODE-based
methods assume intrinsic noise to be negligible is often challenging to justify. The DPP
method imposes multiple model assumptions, such as time invariant rate constants. The SDE-
MEM methods employ approximate simulators to model intrinsic noise, that can be inaccu-
rate when few molecules control the dynamics [13]. Overall, available frameworks only
address specific questions. Moreover, there are scenarios where all existing methods are inade-
quate. For example, when studying a gene expression model with low numbers of molecules

and a time-varying transcription rate, using available inference approaches will impose unrealistic model assumptions, potentially leading to incorrect model predictions [14].

To fully exploit the power, and facilitate usage, of stochastic single-cell dynamic models we propose a flexible Bayesian inference framework for stochastic dynamic mixed-effects models. By building upon a Bayesian inference framework originally thought for SDEMEMs [10], we introduce a novel, computationally efficient inference method that (i) in our experiments has shown to be 35+ times computationally faster than the solution in [10] and (ii) is capable of inferring unknown model parameters when intrinsic noise is modelled by either exact [11, 15], or approximate [12, 13] stochastic simulators. Moreover, by leveraging on the state-of-the-art statistical methods [16, 17], our framework allows for large flexibility in how extrinsic noise is modelled. Using synthetic examples, we show how this flexibility facilitates understanding of a stochastic gene expression model regulated by an extrinsic time-varying signal and cellular pathways where intrinsic noise causes cells to migrate between states. Further, by combining time-lapse microscopy with microfluidics, we employ our inference framework to distinguish between multiple network structures and identify sources of cell heterogeneity in the budding yeast *Saccharomyces cerevisiae* SNF1 nutrient sensing pathway.

## Results

### Inference framework for stochastic dynamic single-cell models

We developed a flexible modelling framework, PEPSDI (**P**articles **E**ngine for **P**opulation **St**ochastic **D**ynam**I**cs), which infers unknown model parameters from dynamic data for single-cell dynamic models that account for both intrinsic and extrinsic noise (Fig 1). The latter can be modelled hierarchically by letting parameters believed to vary between cells (such as protein translation rates) follow a probability distribution. Furthermore, the model parameters can incorporate extrinsic time variant signals, such as the circadian clock [18], and measured extrinsic data, such as cell volume. Intrinsic noise can be modelled by multiple stochastic simulators, specifically the exact Stochastic Simulation Algorithm (SSA, Gillespie) and Extrande simulators [11, 15], and the approximate tau-leaping [13] and Langevin simulators [12]. Hence, PEPSDI is applicable for gene expressions models with low numbers of molecules [19], and signalling models where large numbers of molecules can make exact simulations unfeasible [13]. This framework can further infer the strength of the measurement error, and is suitable when either all, or a subset of the model components are observed. More formally, our methodology produces Bayesian inference for state-space models with latent dynamics incorporating mixed-effects, that is state-space mixed-effects model (SSMEM). It builds upon and expands with increased computational efficiency (see further below), the schemes previously proposed for SDEMEMs [10].

From the statistical inference point of view, PEPSDI is a Gibbs sampler targeting the full posterior distribution of all unknowns. To allow large flexibility in the model construction, some of the Gibbs-steps can be performed using Hamiltonian Monte Carlo (HMC) [17]. For example, it can be assumed that the synthesis and breakdown rates of a protein follow a log-normal distribution and if correlation between the rates is suspected, *a priori*, the HMC sampler permits efficient inference of the correlation [20]. For the Gibbs-steps where the likelihood function is intractable, PEPSDI uses a pseudo-marginal approach employing particle filters [16]. This enables the user to select from a wide range of stochastic simulators [13]. Furthermore, for computational efficiency we employ, when possible, correlated particle filters [21], and tune the parameters proposal distribution using adaptive algorithms [22–24].

Our framework can be run in two ways. The first way employs the Gibbs sampler developed in [10] (Algorithm 2 in S1 Text). As this approach requires a particle filter to estimate the full

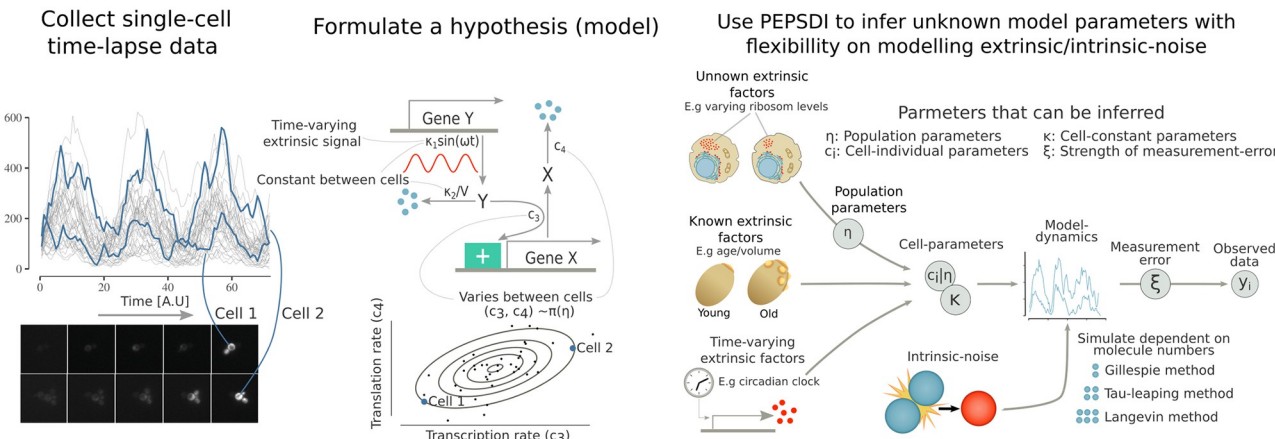

**Fig 1. PEPSDI: A Bayesian inference framework for single-cell stochastic dynamic models.** Single-cell time-lapse data obtained via fluorescent microscopy often exhibits considerable cell-to-cell variability (left). Dynamic modelling (middle) can help elucidate both the reaction dynamics, and sources of cell-to-cell variability behind such data. PEPSDI (right) is a flexible inference framework for dynamic stochastic single-cell models that imposes few model assumptions. For example, extrinsic noise can be modelled by letting cell-individual quantities i) be modelled probabilistically as $\mathbf{c}^{(i)} \sim \pi(\mathbf{c}^{(i)}|\boldsymbol{\eta})$ (unknown extrinsic factors), ii) be combined with known extrinsic data (known extrinsic factors), and iii) be time-variant (time varying extrinsic factors). Furthermore, PEPSDI includes multiple stochastic algorithms for modelling intrinsic noise, and assumes that the observed data $\mathbf{y}^{(i)}$ is acquired with a measurement error. Overall, based on the observed data PEPSDI can infer the cell individual parameters $\mathbf{c}^{(i)}$, the cell constant parameters $\boldsymbol{\kappa}$, the population parameters $\boldsymbol{\eta}$ and the strength of the measurement error $\boldsymbol{\xi}$.

data likelihood (over all individuals) when inferring parameters that are constant between cells, it can be computationally demanding for data sets with many individuals ($> 50$). Hence, we developed a novel Gibbs sampler (Algorithm 3 in S1 Text), where we allow cell constant parameters to vary weakly between cells. This removes the expansive inference of those parameters (full details in Materials and methods and Section A in S1 Text). Our new Gibbs sampler, which is the default option in PEPSDI, can speed up the inference by a factor of 35 for a data set with 100 cells, as measured in wall-clock time compared to the former Gibbs sampler (S3 Fig), and larger speedups are likely attainable when the number of observed cells increases.

Overall, with PEPSDI we extend the Wiqvist's et al. framework, which was conceived for SDEMEMs [10], to a wide array of stochastic dynamic mixed-effects models by leveraging existing particle filters [9, 25] and the flexibility of HMC [17]. PEPSDI is written in Julia [26], and is available on GitHub (https://github.com/cvijoviclab/PEPSDI) (more details in the Materials and methods section). To encourage usage of our framework, the provided code is flexible with regards to the model structure, and all examples are available as notebooks (https://github.com/cvijoviclab/PEPSDI/tree/main/Code/Examples). Guidelines on how to run the inference schemes are presented in Section B in S1 Text, and a tutorial is in Section C in S1 Text. The mathematical description of PEPSDI is provided in Materials and methods and Section A in S1 Text.

## Application to simulated circadian clock gene expression model

We applied the developed inference framework on simulated data from a simple circadian clock [18] gene expression model (Fig 2A). The circadian clock was modelled as a sine function regulating the transcription rate, causing the protein levels to oscillate (Fig 2B). Moreover, to simulate strong intrinsic noise, the numbers of molecules was kept low.

Additional extrinsic noise was simulated by letting the cell-specific rate constants follow a multivariate log-normal distribution, $\mathbf{c}^{(i)} \sim \mathcal{LN}(\boldsymbol{\mu}, \boldsymbol{\Omega})$, where $\boldsymbol{\mu}$ and $\boldsymbol{\Omega}$ are the mean and

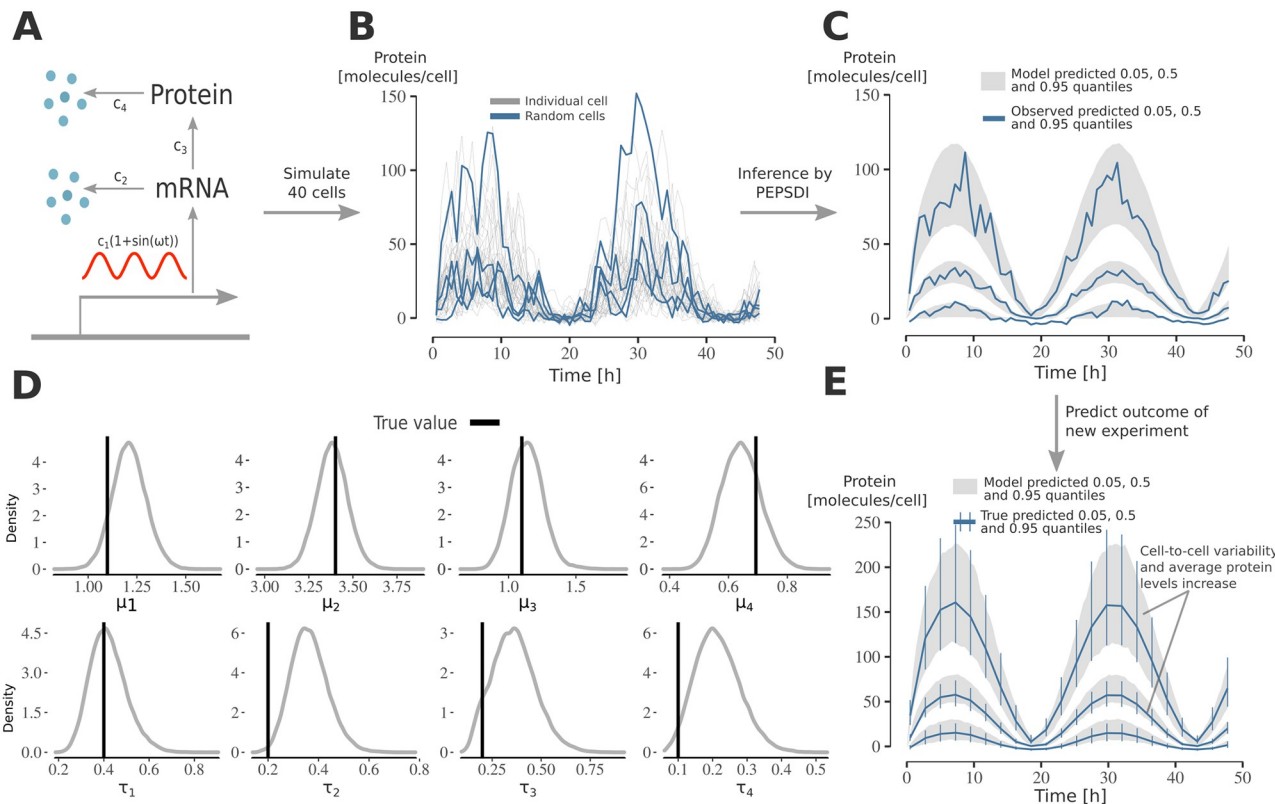

**Fig 2. Applying PEPSDI on synthetic data from circadian clock gene expression model. A)** Schematic representation of the gene expression model. The model consists of two states (mRNA, Protein), and four reactions with associated rate constants $\mathbf{c} = (c_1, \ldots, c_4)$. The circadian clock, modelled as a sine function with a period of 24 hours, regulates the transcriptions activity $c_1$. Extrinsic noise was simulated by assuming that rate-constants jointly follow a log-normal distribution: $\mathbf{c} \sim \mathcal{LN}(\boldsymbol{\mu}, \boldsymbol{\Omega})$. **B)** Protein count per cell for 40 cells simulated using the gene-network model. Data was simulated using the exact Extrande algorithm with an additive Gaussian measurement noise. **C)** Posterior visual checks [30]. The plot was generated as follows: i) from the inferred posterior distribution, simulate 40 cells, and ii) for trajectories corresponding to the 40 cells compute their 0.05, 0.5, 0.95 quantiles at each time point. This was repeated 10, 000 times independently and we then computed the 95% credibility intervals for these quantiles. The blue lines correspond to the observed quantiles (from the data in (B)). **D)** Inference results using the data in B. The plots shows the marginal posterior for $\boldsymbol{\mu} = (\mu_1, \ldots, \mu_4)$ and $\boldsymbol{\tau} = \mathrm{diag}\{\boldsymbol{\Omega}\}^{1/2}$. **E)** Using the inferred model to predict the outcome of adding an extra gene in the model. An extra gene was modelled by doubling the transcription rate $c_1$. The grey lines represent 95% credibility intervals for the 0.05, 0.5 and 0.95 quantiles (obtained as in subplot (C)) when simulating the inferred model with $c_1$ doubled. The blue lines and error bars represent the true 95% credibility intervals when observing 40 cells.

covariance matrix of the associated Gaussian distribution. Its multiplicative nature [27] makes the log-normal distribution a common choice for modelling parameters believed to vary between cells [6]. We also correlated the rate parameters to emulate that protein synthesis and degradation rates might co-vary [5, 28] (additional setup details are in Section D in S1 Text).

Since the standard SSA algorithm cannot handle the time-varying transcription rate, we modelled intrinsic noise via the exact Extrande simulator [15]. Faster approximate algorithms can also be used to model intrinsic noise, but here the low numbers of molecules in the model made these impractical [29].

We ran PEPSDI for 50, 000 iterations and recovered the true model parameters and how they vary within the cell population (Fig 2D and S1 Fig). There is a slight bias in the cell-to-cell variability of the breakdown rates ($\tau_2$, $\tau_4$). When simulating and performing inference on 60 additional cells (thus 100 in total) this bias persists, however the posterior modes get slightly closer to the ground-truth values, compared to when using only 40 cells, and the parameters uncertainty decrease (S1 Fig). This suggests that to fully recover how kinetic rates vary between

cells, if only the protein number is measured, in a simple gene expression model (as in Fig 2A) is challenging.

The inferred model has predictive power. We considered an experiment where an additional gene is inserted into the model, and time-lapse data is collected for 40 new cells. The model accurately predicts an increase in both, protein and cell-to-cell variability levels (Fig 2E).

## Application to simulated stochastic bistable model

Intrinsic noise can have a strong impact on cellular dynamics [31, 32]. Cellular processes have been shown to exhibit stochastic oscillations in gene regulation [31] and stochastic bistability as reported for the *lac*-operon regulation [32]. To study the performance of PEPSDI for such a process, we implemented the Schlögl model (Fig 3A) [33], where cells stochastically migrate between states of high and low gene expression (Fig 3B).

To simulate extrinsic noise, we let one of the synthesis rates, $c_3$, follow a log-normal distribution. To emulate that certain model parameters, such as protein dissociation rates, can have a neglectable variability [4], we kept a synthesis ($c_1$) and dissociation rate ($c_2$) constant between individuals. The synthesis rate $c_4$ was assumed to be known (additional setup details are in Section D in S1 Text).

To account for large numbers of molecules, we modelled intrinsic noise via the fast, approximate Langevin simulator. We then used so-called guided proposals [9], directing simulations towards observed values, which makes the inference more efficient for models with stochastic events.

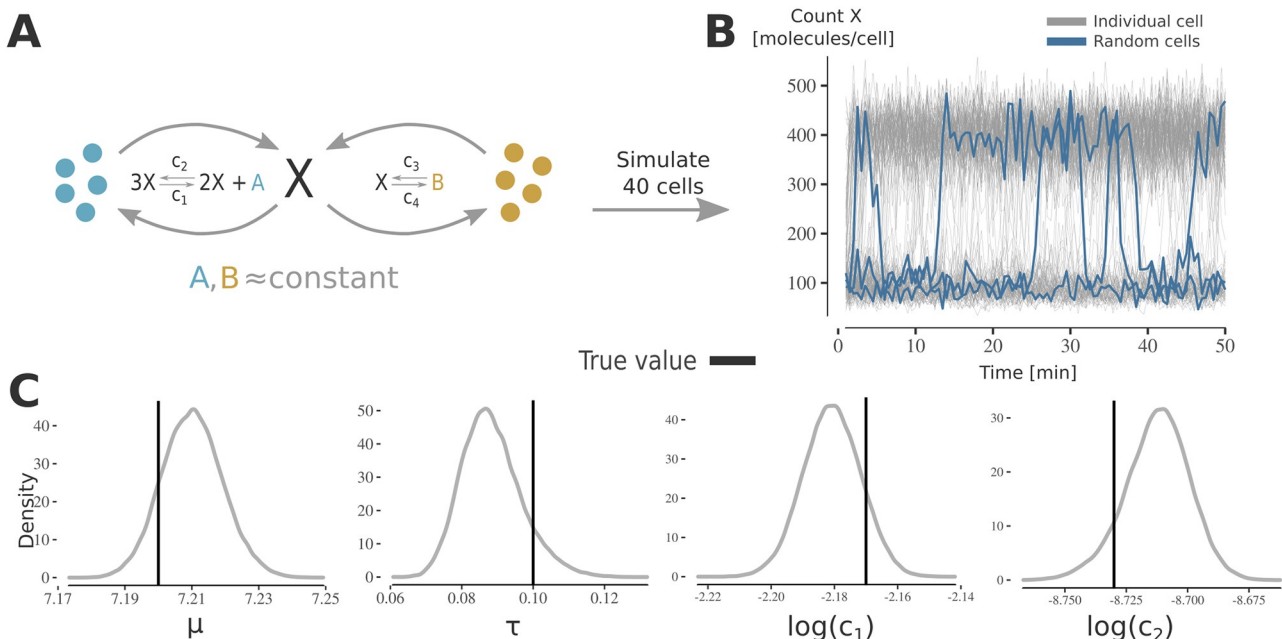

**Fig 3. Applying PEPSDI on synthetic data from stochastic bi-stable model. A)** Schematic representation of the stochastic bi-stable Schlögl model. Since the species ($A$, $B$) are assumed to be available in excess, the model consists of one state ($X$) and four reactions with associated rate constants $\mathbf{c}$ = ($c_1, \ldots, c_4$). Extrinsic noise was simulated by assuming that $c_1$ follows a log-normal distribution; $c_1 \sim \mathcal{LN}(\mu, \tau^2)$. The remaining parameters are assumed to be cell-constant and $c_4$ is assumed to be known. **B)** Molecule count of $X$ per cell for 150 cells simulated using the Schlögl model. Data was simulated using the SSA algorithm with an additive Gaussian measurement noise. Noticeably, a subset of cells stochastically migrates between two cell-states. **C)** Inference results using the data in (B)). To efficiently infer the posterior distribution the model was simulated using the Langevin simulator. The plots show the marginal posterior for the population parameters ($\mu$, $\tau$), and the cell-constant parameters ($c_2$, $c_3$).

We ran PEPSDI for 100, 000 iterations, and posterior distributions of the modelled quantities were recovered (Fig 3C and S2 Fig), allowing for better understanding of mechanistic properties of the model. For example, by simulating the inferred model it becomes apparent that low values of the synthesis rate $c_1$ commit cells to a first cell-state, e.g of low gene-expression, while for large values of $c_1$, cells commit to, and jump between, two different states of gene expression (S2 Fig).

### Performance evaluation of adaptive MCMC proposals

To propose unknown model quantities that rely on a pseudo-marginal step in our Gibbs sampler (see Materials and methods), PEPSDI includes three adaptive Markov chain Monte Carlo (MCMC) schemes: the adaptive metropolis (AM) [22], the AM with global scaling (Alg. 4 in [23]), and the robust AM (RAM) samplers [24]. These schemes were developed for problems where the likelihood function is available, not approximated via a Monte Carlo scheme as in the pseudo-marginal methods. Thus, we benchmark these three adaptive MCMC proposal schemes to see how they perform with stochastic approximations to the likelihood (full details in Section D in S1 Text). For computational reasons, single time-series inference for two different models is considered.

For the Schlögl model (Fig 3A), the stochastic bistability (Fig 3B) causes the likelihood approximation used in the pseudo-marginal method to have a large variance despite the usage of guided proposals. To investigate if this impacts the performance of adaptive MCMC schemes, we launched multiple inference runs with 60,000 iterations each. Overall, the RAM sampler had on average the highest multiple effective sample size (MultiESS) value [34], thus providing a larger number of nearly independent samples (Fig 4A). This sampler also had the smallest variability in the MultiESS, suggesting it is robust against, for example, bad starting guesses (start guess 3 and 4 in Fig 4A).

For the Ornstein-Uhlenbeck stochastic differential equation model (Section D in S1 Text), the likelihood approximation has a small variance and exact Bayesian inference is possible because the likelihood can be exactly calculated using the Kalman filter [35]. Hence, in addition to the same setup as for the Schlögl model, we compared the several posteriors obtained via particles-based PEPSDI methodology (by using different number of particles and different starting parameters) against the "gold standard" posterior produced with the exact likelihood. The comparison was produced by computing the first order Wasserstein distance between each PEPSDI posterior and the exact one for the three kinetic parameters in the model. We used the last 15,000 posterior samples to compute the distance via the R `transport` package [36]. Overall, the RAM sampler has the best MultiESS (Fig 4B), and the Wasserstein distance for the RAM sampler is the smallest and has the smallest variability (Fig 4C).

### Application to glucose repression pathway

After considering synthetic examples, we set out to study the SNF1 pathway in *S.cerevisiae*. The SNF1 complex, and its mammalian homolog AMPK plays a major role in both metabolic regulation and maintenance of cellular homeostasis. In response to stress, such as ageing or nutrients limitation, SNF1 mediates the signal transduction to transcription factors. Mig1 is a transcriptional repressor which the SNF1 complex deactivates when energy-rich carbon sources are limited. This is followed by Mig1 relocalisation to the cytoplasm and release of repression of genes responsible for utilisation of alternative carbon sources [37, 38]. If the amount of energy-rich nutrients is elevated, Mig1 translocates to the nucleus [39]. This process is accompanied by Mig1 dephosphorylation where the Reg1 phosphatase plays the main role

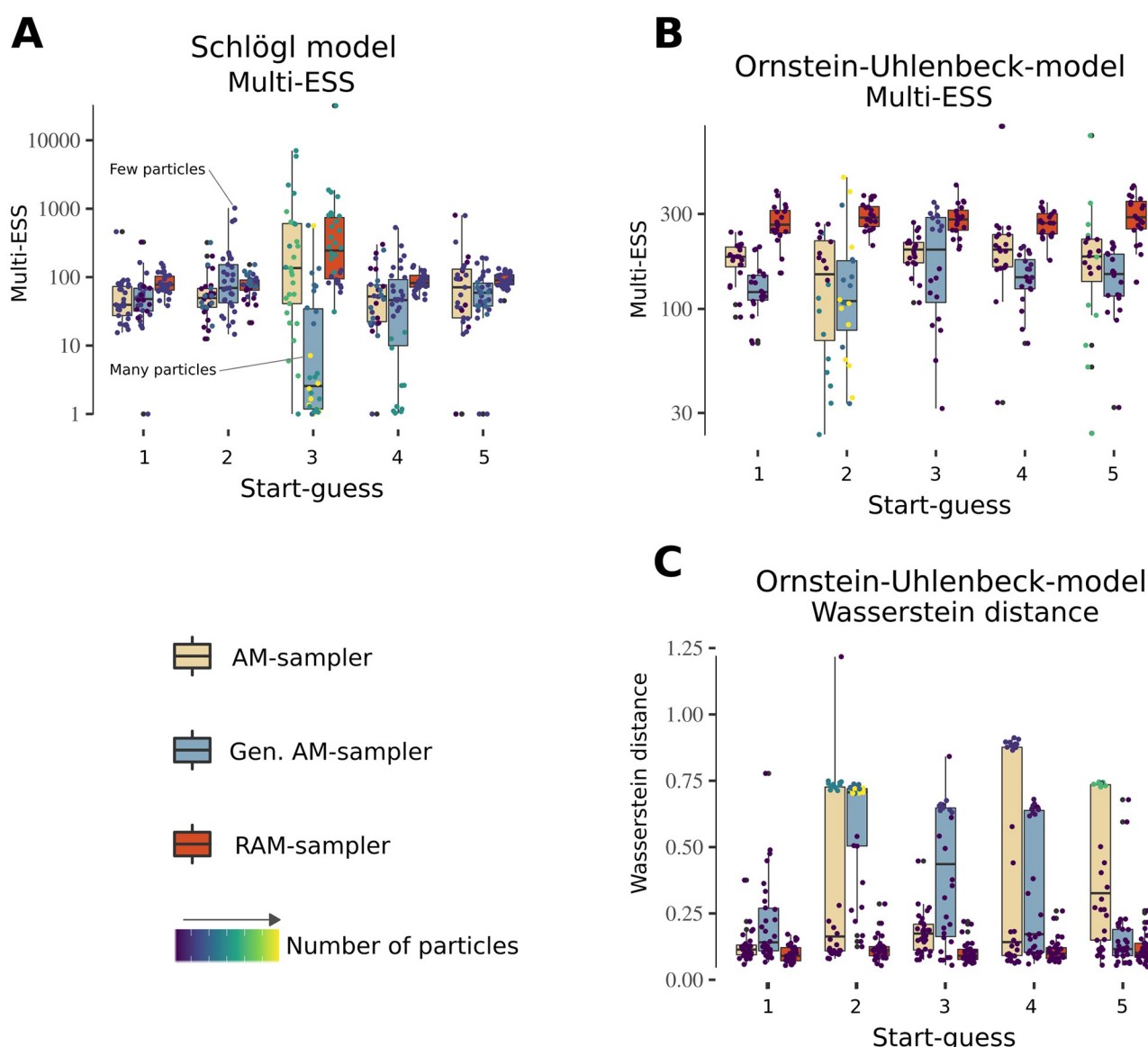

**Fig 4. Benchmarking adaptive MCMC-proposals for pseudo-marginal inference. A**) Benchmark results for the Schlögl model. The quality of the adaptive schemes was measured using the MultiESS-criterium [34], where higher values are better. For computational reasons the benchmark was performed for a single-individual (single time-series data). Overall, we simulated three datasets. For each dataset we ran five pilot runs with initial parameter values set at randomly chosen prior locations, and then tuned the number of particles (Section D in S1 Text). Starting from the last drawn parameter value in each pilot run, 10 further inference runs each of 60, 000 iterations were independently launched. For each chain, we then discarded the 20% first iterations, and used the remaining samples to compute the MultiESS (of all inferred parameters). The colours denote the several adaptive proposal schemes, and the colour bar represents the number of particles selected by the tuning scheme after the pilot run. A high number of particles implies longer run-times, and an inefficient pilot run. **B-C**) Results for the MultiESS and Wasserstein distance for the Ornstein-Uhlenbeck model. The benchmark conditions were the same as in A). The Wasserstein distance was approximated by using the last 15,000 posterior samplers of each chain for the kinetic parameters ($c_1, c_2, c_3$), and compared against the last 15,000 samples from a gold-standard posterior inference run where the Kalman filter was used to exactly compute the likelihood of the Ornstein-Uhlenbeck model.

[1]. While this signalling cascade is well studied, it has also been indicated that the SNF1 pathway dynamics exhibits a high range of cell-to-cell variability [40].

To deduce the dynamics of the SNF1 pathway at a cell level, time-resolved data is required. We utilised fluorescent microscopy to follow Mig1 on a single-cell level and observe its localisation over time (Fig 5A and 5B). This was coupled with microfluidics systems enabling high

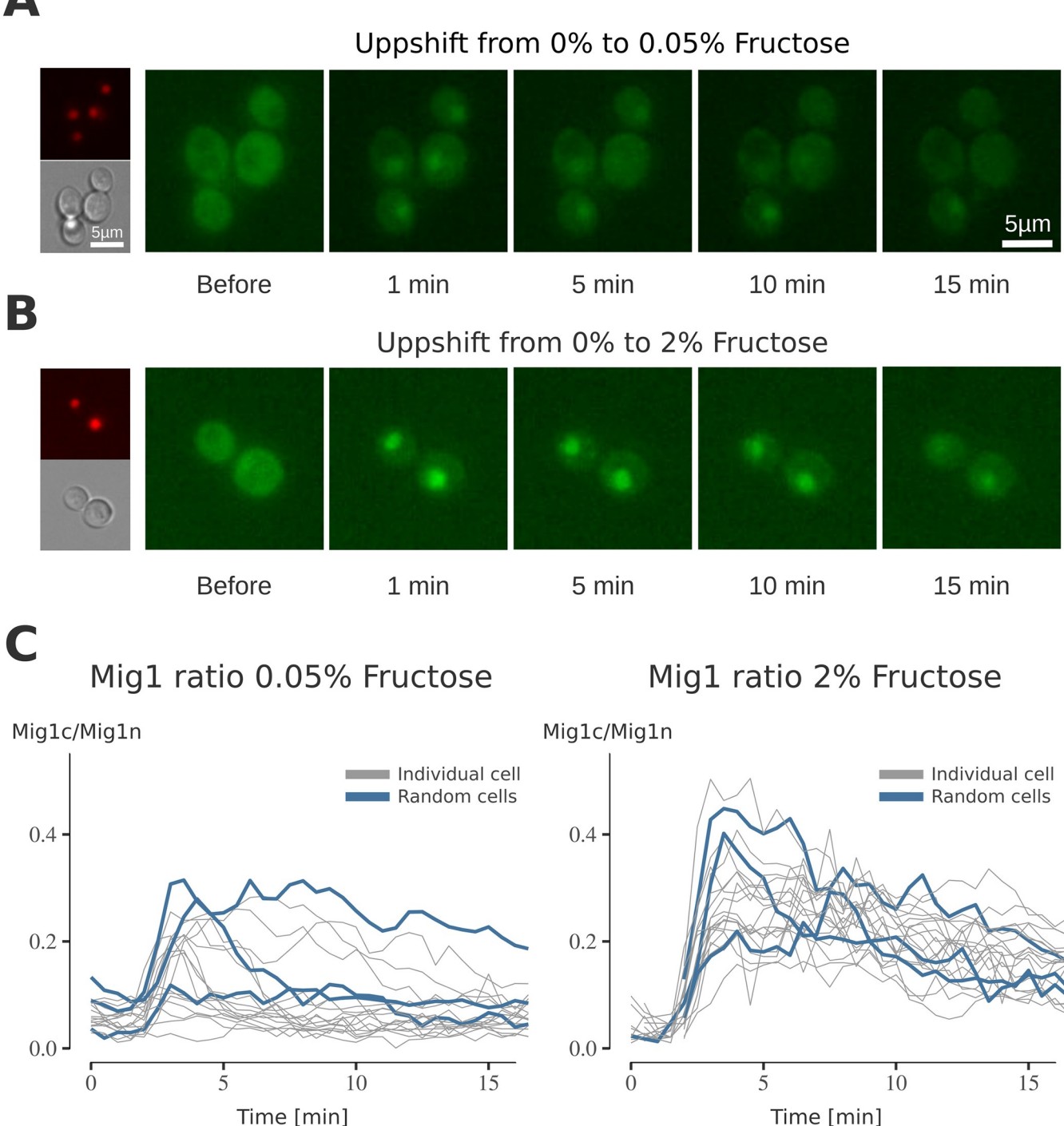

**Fig 5. The impact of fructose on Mig1 dynamics.** Mig1 localisation in response to the media exchange from media containing 0% fructose (no fast-fermentable carbon sources present) to media containing **A)** 0.05% fructose and **B)** 2% fructose. The white lines correspond to $5\mu m$ scale bars. Exchange of media was achieved through an open microfluidic system. Green fluorescent protein (GFP) images depict Mig1 localisation before and after switching of the media at the noted times. Brightfield images taken as control for the cell localisation. Red fluorescent protein (RFP) images depict Nrd1, a protein which is stationary in the nucleus and used as nuclear marker. **C)** The nuclear intensity of Mig1 for each single cell in the experiment is given by the localisation index of Mig1 over time (minutes). Localisation index is determined by (Mig1n-Mig1c)/Mig1c (for short called Mig1n / Mig1c in the paper) with Mig1n being the intensity of Mig1 in the nucleus and Mig1c the intensity in the complete cell. All cell traces are grey, three random selected cells are given in blue. Combined data sets consist of $N = 37$ individuals, with $N_{0.05} = 22$ for the 0.05% and $N_{2.0} = 15$ cell for the 0.05% and 2.0% fructose experiments, respectively.

control of the cell environment. Mig1 localisation was followed during 15 min after cells were switched from carbon source-depleted conditions to two different fructose concentrations (w/v): 0.05%, and 2% (Fig 5C).

## Modelling of Mig1 nuclear dynamics

To elucidate both the source of cell heterogeneity and reaction mechanisms behind the Mig1 dynamics data (Fig 5C), we formulated and analysed two network structures (Fig 6A and 6B) and two extrinsic noise sources (Fig 6C and 6D), resulting in four plausible models. In both network structures, Mig1 shuttles between the cytosol and nucleus in a carbon source-dependent [41] and -independent manner [42]. The carbon source-dependent response has been observed to occur in two phases: a transient initial Mig1 nuclear entry followed by nucleocytoplasmic shuttling [43]. This can be explained by two independent pathways regulating glucose derepression. One pathway activating the Snf1 kinase and the other one directing Snf1 towards Mig1 [44]. We investigated whether the first pathway promotes Mig1 nuclear entry via a fast signal (Fig 6A), or if Mig1 nuclear entry is delayed due to Reg1 activation (Fig 6B). The second pathway was modelled via an unknown metabolic component since Mig1 dynamics are closely intertwined with metabolic activity [40, 43] (Section E in S1 Text).

Multiple extrinsic noise sources likely affect Mig1 shuttling dynamics. Upstream, Mig1 is regulated by the metabolic activity [43], which in turn is affected by noise sources, such as cell cycle state, cell-lineage effects like cell wall composition, and cell varying protein levels. Since our experimental setup could not distinguish between these sources we lump and model them by letting the rate constants ($c_1$, $c_2$) vary between cells (Fig 6C), as the latter incorporate multiple processes in the initial glycolysis. Since the rates ($c_1$, $c_2$) also encompass fructose abundance, we assume and infer fructose-dependent log-mean values. The second extrinsic noise source proposal (Fig 6D) takes into account that the cell-to-cell variability might be fructose-dependent. To capture extrinsic noise arising from varying protein levels, we assumed that the Mig1 initial values vary between cells ($Mig1c_{t0}$, $Mig1n_{t0}$).

We employed literature-supported priors (Section E in S1 Text) and ran PEPSDI multiple times for each proposed model. Models were compared using posterior visuals check [30], that has the capability to capture both the observed trend and cell-to-cell variability (Fig 6E and 6F and S4 Fig). Overall, the model with a delayed fructose activation of Mig1 nuclear export and a fructose-dependent cell-to-cell variability in Mig1 regulation best described the data (model 2B Fig 6F).

## Cell-to-cell variability in Mig1 nuclear dynamics is tightly regulated by fructose availability

After selecting the best model, we set out to investigate the characteristics of the inferred model parameters. To be interpretable a parameter should be inferred unambiguously, i.e., be identifiable. Assessing identifiability rigorously for a mixed-effects models with stochastic dynamics is outside the scope of this paper, however we addressed identifiability by comparing the posterior distributions from 8 different inference runs. For some parameters, like log-mean($c_1$), the runs converged to different modes showing that parameters cannot be determined uniquely (S4 Fig). However, some parameter properties were consistent between runs, like a strong correlation between ($c_1$, $c_2$) (Fig 6H and S4 Fig) suggesting that certain properties are identifiable.

The inferred coefficient of variations for the rates that describes upstream metabolic regulation of Mig1 localisation ($c_1$, $c_2$), shows that these rates substantially differ between cells (Fig 6H and S4 Fig). When the model is fitted without variability in these rates, that is cell-to-cell

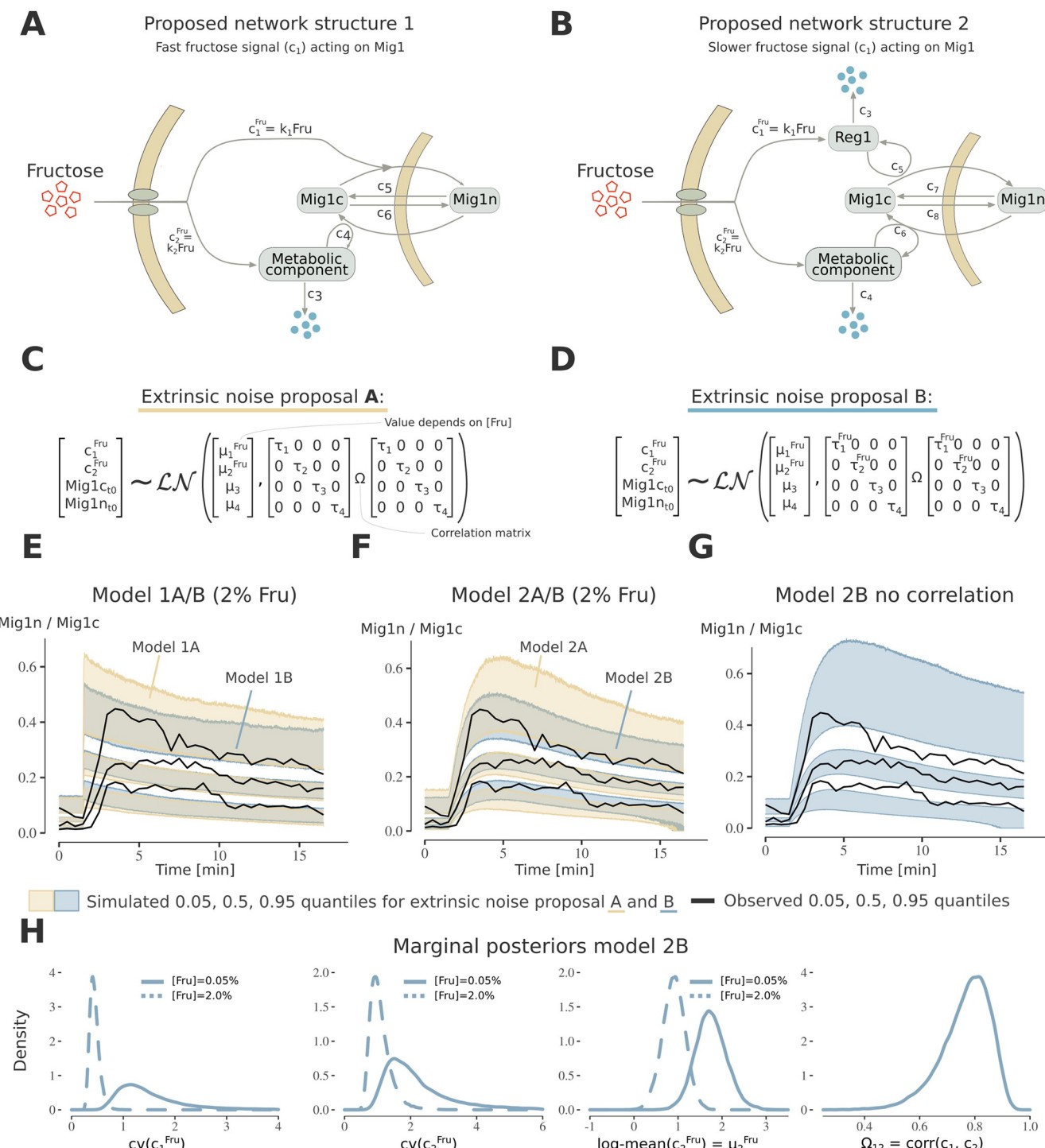

**Fig 6. Modelling Mig1 dynamics in response to fructose addition. A)** Proposed network structure 1 of Mig1 localisation dynamics. The model consists of three states, nuclear Mig1 (Mig1n), cytosolic Mig1 (Mig1c) and a metabolic component. The model proposes that a fructose signal directs Mig1 nuclear import, and that nuclear export is directed by a metabolic signal. **B)** Proposed network structure 2. In addition to structure 1 Mig1 nuclear entry is modelled as delayed due to Reg1 activation. **C)** Modelling proposal A of extrinsic noise. Since $(c_1, c_2)$ encompass multiple cellular processes, and initial Mig1-levels are cell-varying, extrinsic noise was modelled by letting these follow a full log-normal distribution. Since $(c_1, c_2)$ include the external fructose we model different log-mean values for the 2% and 0.05% conditions. **D)** Extrinsic noise proposal B. Additionally to proposal A, we model fructose dependent variability ($\tau$-values) for $(c_1, c_2)$. **E-F)** Posterior visual check for the 2% fructose data for model structure 1 and 2, using extrinsic noise-proposal A and B. The credibility intervals (bands) were obtained as in Fig 2. Black lines are the observed quantiles. Model structure 2 with extrinsic noise proposal B (model 2B) best describes the observed trend, and cell heterogeneity. **g)** Posterior visual check as in F) for model 2B with no correlation between $(c_1, c_2, Mig1n, Mig1c)$ (diagonal $\mathbf{\Omega}$). This yields an increase in cell-to-cell variability seen by the 0.05, and 0.95 quantile credibility intervals. **H)** Marginal posterior distribution for a subset of parameters in model 2B. The log-normal coefficient of variations $cv(c_i) = \exp(\tau_i^2) - 1$ are fructose dependent, and $(c_1, c_2)$ are strongly correlated.

variability is assumed to only arises from varying Mig1 levels and intrinsic noise, the model does not capture the observed cell heterogeneity (S4 Fig). Taken together, this suggests that upstream extrinsic noise contributes strongly to cell-to-cell variability in Mig1 localisation dynamics. Furthermore, the magnitude of the variation in this extrinsic noise, particularly the rate of activation of Reg1 via fructose ($c_1$), is larger for low fructose conditions.

Our results show a correlation in the upstream extrinsic noise sources, specifically the ($c_1$, $c_2$) rates (Fig 6H and S4 Fig), implying that cells with a strong fructose-dependent nuclear import of Mig1, also have a stronger nuclear export. By simulating the model without correlation, we confirm that this correlation regulates cell-to-cell variability (Fig 6G). Thus, our result suggests that co-regulation of two fructose-dependent pathways controls cell heterogeneity in Mig1 localisation dynamics.

The log-mean value of the rate parameter activating the metabolic component (log-mean ($c_2$)) shows that the magnitude of the long-term nuclear export of Mig1 is weaker in high fructose (Fig 6H and S4 Fig). This is consistent with previous reports that nuclear export of Mig1 is primarily an effect of sugar depletion [41].

## The hexokinase Hxk1 is a source of cell heterogeneity in Mig1 localisation dynamics

In model simulations, the cell-varying rate constant $c_1$ linearly correlates with the short-term (0–15 min) Mig1 localisation upon fructose addition to sugar-starved cells (S5 Fig). Thus, since $c_1$ captures a signalling process acting on Reg1 from the initial hexose metabolism via the hexokinase Hxk1, this result suggests a relationship between Hxk1 and observed cell-to-cell variability.

To validate this prediction, we collected single-cell time-lapse microscopy data from a *hxk1Δ hxk2Δ* strain carrying Hxk1-expressing plasmids where both Mig1 localisation and Hxk1 expression are monitored upon fructose addition. In line with model predictions, the observed Hxk1 expression linearly correlates with Mig1 localisation, and the magnitude of variability in localisation explained by Hxk1 (linear regression $R^2$-value) matches the predicted magnitude (S5 Fig). We hence conclude Hxk1 as a source of cell heterogeneity in Mig1 dynamics.

## Discussion

Understanding the inherited nature of how biological processes dynamically change over time, and exhibit intra- and inter-individual variability, has been a major focus of systems biology. The rise of single-cell fluorescent microscopy has enabled the study of those phenomena, but further progress is limited by the availability of methods that facilitate modelling, the essential follow up for rationalisation of such data. To address this, we developed PEPSDI, a versatile Bayesian inference framework that (i) makes it possible to handle a wide range of mixed-effects models with stochastic dynamics by appropriately exploiting existing algorithms, and (ii) introduces a novel inference algorithm that makes computations considerably more scalable, thus opening up for the possibility to model hundreds cells at once (full mathematical description in Section A in S1 Text). We used this algorithm to recover true model quantities for a circadian clock stochastic gene expression model, and deduce mechanistic details for a model where cells stochastically move between two states of gene expression. We also studied SNF1 signalling in yeast and identified hexokinase activity as a source of extrinsic noise, and deduced that sugar availability dictates cell-to-cell variability.

Modelling of Mig1 dynamics (Fig 6) suggests larger cell heterogeneity in upstream extrinsic noise, specifically in the fructose-dependent activation of Reg1 ($c_1$) and the component that

regulates Mig1 shuttling ($c_2$), upon fructose limitation (Fig 6H). We hypothesise the presence of a sugar-dependent biological switch, triggered when fructose is present. This is consistent with the fact that under low fructose, only a few cells due to extrinsic noise trigger the switch leading to larger cell-to-cell variability. Moreover, results show that the magnitude of $c_2$ is larger under fructose limitation (Fig 6H). It has been reported that rapid activation of the SNF1 complex upon glucose starvation is ensured by increased levels of ADP [45]. We therefore suggest that within the SNF1 pathway, ADP is a central part of the $c_2$ rate constant, facilitating nuclear export of Mig1 via activation of the SNF1 complex. This is consistent with elevated ADP levels upon limited glucose or fructose concentrations in the cellular environment. Moreover, hexokinases that we identified to be a part of the $c_1$ rate, regulate Reg1, which in turn also requires the abundant presence of a hexose sugar for full activity of the phosphatase [46]. Taken together, we hypothesise that $c_1$ and $c_2$ incorporate hexokinases and ADP, respectively, and both are controlled via metabolism. This creates a tight co-regulation of the SNF1 pathway, which controls cell heterogeneity (Fig 6G).

Besides studying Mig1 reaction dynamics, we used our framework to investigate noise sources behind the observed cell heterogeneity. Our results suggest that upstream extrinsic noise arising from metabolic activity is a major source of cell variability in Mig1 regulation (S5 Fig). Beyond this work, dynamic modelling combined with single-cell high quality time-lapse data has been used to elucidate sources of noise in gene expression [4] and to understand the role of extrinsic factors, like cell age, on gene expression [5]. As collecting time-lapse data often is easier than directly measuring sources of cellular noise [8, 47, 48], PEPSDI is likely one of the most promising tools to untangle it. Being more flexible than previously proposed frameworks, we believe that our approach can be applied to many more systems beyond Mig1 signalling, where noise plays an important role.

The reason PEPSDI is flexible rests on its modifiable nature. This modularity facilitates modelling of intrinsic noise by either the SSA [11], Extrande [15], tau-leaping [49] or Langevin [12] stochastic simulators. Additionally, new modules, such as the hybrid-simulators [50] used to study NF$\kappa$B-pathway [51], can be incorporated. Likewise, new particle filters for the pseudo-marginal modules can be added. Guided particle filters, like the one we used for the Schlögl model, are particularly statistically efficient (as described in Section A in S1 Text) [9]. However, most guided filters are restricted to observational models having a linear structure with additive Gaussian measurement noise, which is not always applicable for data in systems biology and medicine [7]. The development of more flexible guided filters is thus important to make frameworks like ours more efficient for a wide range of models.

Reproducibility and usability are key concepts for ensuring advances in our understanding of biological processes. Besides being modifiable, we aimed to make our framework accessible, by providing extensive tutorial notebooks on both, how to use it and how to leverage the underlying pseudo-marginal framework to model single time series data. Additionally, to help users of pseudo-marginal inference, we evaluated adaptive Markov chain Monte Carlo (MCMC) proposals [22–24], which resulted in the RAM sampler [24] displaying the best performance (Fig 4). However, our results were based on two models, and these also depend on the specific parameters we used for each adaptive scheme (Section D in S1 Text). Further analysis is thus required before generalising our conclusions.

In summary, we have developed and employed a framework to deduce the reaction dynamics, and sources of cell heterogeneity behind single-cell time-lapse data. Since PEPSDI is an inference framework for dynamic state-space mixed-effects models, it can also be applied for problems arising in ecology [9], neuroscience [10] and in pharmacokinetics and pharmacodynamics (PKPD) [35]. Considering that PEPSDI does not impose strict model assumptions, it is easy to envision additional applications, such as modelling the behaviour of cancer cell

populations [52]. We believe that this approach will play an increasingly important role in addressing challenging biological questions that cannot be answered by experimental approaches only, thus providing novel insights for better understanding life processes.

## Materials and methods

### Stochastic simulations

Under the assumption that the system is well mixed, the dynamics of a stochastic reaction-network of $d$ species, $\mathbf{X} = (X_1(t), \ldots, X_d(t))$, is described by the chemical master equation [13, 49]. The propensity function $h(\mathbf{x}, \mathbf{c}, t)$ is a measure of the probability that one, out of $R$, reactions will occur depending on $\mathbf{X}$ and the rate-constants $\mathbf{c}$. Considering that the master-equation can rarely be solved due to its probabilistic nature [13, 49], our inference framework relies on simulating from it.

The SSA direct method [11] produces exact stochastic simulations. Similarly, the Extrande simulator [15] produces exact solutions when the rate-constants $\mathbf{c}$ are time variant. However, both approaches simulate each reaction event and thus can be slow for large propensities [53]. Assuming that the propensities do not change noticeably during the time interval $[t, t + \tau]$ (leap condition 1), the reactions will be close to independence of each other. Hence, the tau-leap approach with fixed step-length can be used to update the state vector from time $t$ to $t + \tau$ [12, 54]. Further assuming that the propensities are sufficiently large in $[t, t + \tau]$ (leap conditions 2), the state-vector can be updated via the chemical Langevin stochastic differential equation [12, 49]. Typically, the number of molecules must be large for leap condition 1 and 2 to hold.

Our framework allows models to use the SSA method and the Extrande method when neither leap conditions holds. When one or both conditions holds, our framework allows usage of either tau-leaping or the Langevin approach.

### PEPSDI: Bayesian inference for single-cell dynamic models

PEPSDI performs Bayesian inference for state-space models with latent dynamics incorporating mixed-effects, shortly "state-space mixed-effects models" (SSMEMs). A state-space model is a discrete-time, stochastic model that contains two sets of equations: (i) one describing how a latent Markov process transitions in time (the state equation) and (ii) another one describing how an observer measures the latent process at each discrete time-point (the observation equation), assuming conditional independence between observations given latent states. Thus, a state-space model can describe a stochastic chemical reaction network. Here we outline PEPSDI, and a more complete description also showing pseudo-algorithms and our novel and efficient Gibbs sampler, is provided in Section A in S1 Text.

To perform inference, PEPSDI requires single-cell time-lapse measurements $\mathbf{y}_l^{(i)}$ for the $i$-th individual collected at $(l = 1, \ldots, n_i)$ discrete time-points $t^{(i)} = (t_1^{(i)}, \ldots, t_{n_i}^{(i)})$ for $i = 1, \ldots, M$ individuals. Note, from now we denote with "individual" the measurements from a single cell. However, since PEPSDI can be used in other applied areas (e.g. ecology, PKPD, etc), an individual is more generally a unit from the population of interest. The individual data is assumed to be noise-corrupted as $\mathbf{y}_l^{(i)} = \mathbf{g}(\mathbf{x}_l^{(i)}, \boldsymbol{\epsilon}_l^{(i)})$, where we use the shorthand notation $z_l^{(i)}$ to denote a variable $z$ observed at time $t_l^{(i)}$. Here, $\boldsymbol{\epsilon}_l^{(i)}$ can be considered as unobservable noise (e.g. measurement error) which follows an error distribution $\boldsymbol{\epsilon}_l^{(i)} \sim \pi_{\boldsymbol{\epsilon}}(\xi)$, and $\mathbf{g}(\cdot)$ is a (possibly nonlinear) function of its arguments. To infer parameters in the nutrient-sensing Mig1 pathway, the observed Mig1 data represents a ratio of nuclear to cytosolic intensity, and thus

$g(\mathbf{x}_l^{(i)}, \epsilon_l^{(i)}) = Mig1N(t)/Mig1C(t) + \epsilon_l^{(i)}$, with independent $\epsilon_l^{(i)} \sim \mathcal{N}(0, \sigma^2)$. We assumed this error model since it has shown to work well in other modelling studies working with single-cell fluorescent data for the SNF1 pathway [28, 40, 55].

For a SSMEM, PEPSDI infers (i) the vector of individual rate-constants $\mathbf{c} = (\mathbf{c}^{(1)}, \ldots, \mathbf{c}^{(M)})$ where we assume $\mathbf{c}^{(i)} \sim \pi(\mathbf{c}|\boldsymbol{\eta})$ $(i = 1, \ldots, M)$, (ii) cell-constant rate-parameters $\boldsymbol{\kappa}$ that are assumed to be shared by all cells, (iii) the parameters $\boldsymbol{\xi}$ for the measurement error, and (iv) the population parameters $\boldsymbol{\eta}$. The population parameters describe the distribution of the individual parameters, $\mathbf{c}^{(i)} \sim \pi(\mathbf{c}|\boldsymbol{\eta})$. For example if $\mathbf{c}^{(i)}$ follows a log-normal distribution, the population parameter corresponds to $\boldsymbol{\eta} = (\boldsymbol{\mu}, \boldsymbol{\Omega})$ and individual rate constants to $\mathbf{c}^{(i)} \sim \mathcal{LN}(\boldsymbol{\mu}, \boldsymbol{\Omega})$. By stacking the measurements from all individuals into $\mathbf{y} = (\mathbf{y}^{(1)}, \ldots, \mathbf{y}^{(M)})$, the posterior distribution we target is:

$$\pi(\mathbf{c}, \boldsymbol{\kappa}, \boldsymbol{\eta}, \boldsymbol{\xi}|\mathbf{y}) \propto \pi(\mathbf{c}^{(1)}, \ldots, \mathbf{c}^{(M)}, \boldsymbol{\kappa}, \boldsymbol{\eta}, \boldsymbol{\xi}) \prod_{i=1}^{M} p(\mathbf{y}^{(i)}|\mathbf{c}^{(i)}, \boldsymbol{\kappa}, \boldsymbol{\xi}), \tag{1}$$

where $\pi(\mathbf{c}^{(1)}, \ldots, \mathbf{c}^{(M)}, \boldsymbol{\kappa}, \boldsymbol{\eta}, \boldsymbol{\xi})$ is the joint prior and $p(\mathbf{y}^{(i)}|\mathbf{c}^{(i)}, \boldsymbol{\kappa}, \boldsymbol{\xi})$ is the likelihood term for individual $i$. Note, in Eq 1 we have assumed that measurements from different individuals are conditionally independent, given the individual-specific $\mathbf{c}^{(i)}$ and the population parameters.

The posterior (Eq 1) is high-dimensional, and ideally we could sample from it via a Gibbs-sampler [56] by looping through the following three steps:

$$1. \ \pi(\mathbf{c}|\boldsymbol{\kappa}, \boldsymbol{\eta}, \boldsymbol{\xi}, \mathbf{y}) \propto \prod_{i=1}^{M} \pi(\mathbf{c}^{(i)}|\boldsymbol{\eta}) \pi(\mathbf{y}^{(i)}|\mathbf{c}^{(i)}, \boldsymbol{\kappa}, \boldsymbol{\xi})$$

$$2. \ \pi(\boldsymbol{\kappa}, \boldsymbol{\xi}|\mathbf{c}, \boldsymbol{\eta}, \mathbf{y}) \propto \pi(\boldsymbol{\kappa}, \boldsymbol{\xi}) \prod_{i=1}^{M} \pi(\mathbf{y}^{(i)}|\mathbf{c}^{(i)}, \boldsymbol{\kappa}, \boldsymbol{\xi}) \tag{2}$$

$$3. \ \pi(\boldsymbol{\eta}|\mathbf{c}, \boldsymbol{\kappa}, \boldsymbol{\xi}, \mathbf{y}) \propto \pi(\boldsymbol{\eta}) \prod_{i=1}^{M} \pi(\mathbf{c}^{(i)}|\boldsymbol{\eta}).$$

Notice that it is possible to sample from the first step by independently sampling for each of the $\mathbf{c}^{(i)}$ separately from the other ones. That is, step 1 can be written as

$$1. \ \pi(\mathbf{c}^{(i)}|\boldsymbol{\kappa}, \boldsymbol{\eta}, \boldsymbol{\xi}, \mathbf{y}) \propto \pi(\mathbf{c}^{(i)}|\boldsymbol{\eta}) \pi(\mathbf{y}^{(i)}|\mathbf{c}^{(i)}, \boldsymbol{\kappa}, \boldsymbol{\xi}), \qquad i = 1, ..., M \tag{3}$$

and hence the sampling step for each $\mathbf{c}^{(i)}$ only needs to access the corresponding individual-specific $\pi(\mathbf{y}^{(i)}|\mathbf{c}^{(i)}, \boldsymbol{\kappa}, \boldsymbol{\xi})$.

However, in practice, steps 1–2 cannot trivially be sampled from, due to the intractability of the likelihood for the $i$-th individual $\pi(\mathbf{y}^{(i)}|\boldsymbol{\kappa}, \boldsymbol{\xi}, \mathbf{c}^{(i)})$ which is defined by a multidimensional integral (a precise expression is given in Section A in S1 Text), and here sampling is performed using a pseudo-marginal approach following [10]. The posterior targeted in step 3 is tractable, and thus $\boldsymbol{\eta}$ is sampled using Hamiltonian Monte Carlo [17, 57, 58].

PEPSDI can be run with two flavours. Both sample the conditionals in Eq 2 via, when required, pseudo-marginal approaches. However, the default option is to slightly perturb the SSMEM. This prevents the need of step 2 in the Gibbs-sampler, resulting in substantially shorter run-time. To properly motivate this perturbation, we first cover pseudo-marginal particles-based inference.

## Pseudo-Marginal particles-based inference

The pseudo-marginal Metropolis-Hastings scheme samples from the desired posterior by considering the marginal of an augmented posterior [16, 59]. More details are given in Section A in S1 Text. Briefly, the pseudo-marginal approach considers

$$\pi(\mathbf{c}^{(i)}, \boldsymbol{\kappa}, \boldsymbol{\eta}, \boldsymbol{\xi}, \mathbf{u}^{(i)}|\mathbf{y}^{(i)}) \propto \pi(\boldsymbol{\kappa}, \boldsymbol{\eta}, \boldsymbol{\xi})\pi(\mathbf{c}^{(i)}|\boldsymbol{\eta})\hat{\pi}_{\mathbf{u}^{(i)}}(\mathbf{y}^{(i)}|\mathbf{c}^{(i)}, \boldsymbol{\kappa}, \boldsymbol{\xi})\pi(\mathbf{u}^{(i)}) \tag{4}$$

where often (though not necessarily) parameters can be a-priori independent $\pi(\boldsymbol{\kappa}, \boldsymbol{\eta}, \boldsymbol{\xi}) = \pi(\boldsymbol{\kappa})$ $\pi(\boldsymbol{\eta})\pi(\boldsymbol{\xi})$, and the $\mathbf{u}^{(i)} \sim \pi(\mathbf{u}^{(i)})$ are auxiliary variables used to obtain an unbiased estimate $\hat{\pi}_{\mathbf{u}^{(i)}}(\mathbf{y}^{(i)}|\mathbf{c}^{(i)}, \boldsymbol{\kappa}, \boldsymbol{\xi})$ of $\pi(\mathbf{y}^{(i)}|\mathbf{c}^{(i)}, \boldsymbol{\kappa}, \boldsymbol{\xi})$, that is

$$\mathbb{E}_{\mathbf{u}^{(i)}}[\hat{\pi}_{\mathbf{u}^{(i)}}(\mathbf{y}^{(i)}|\mathbf{c}^{(i)}, \boldsymbol{\kappa}, \boldsymbol{\xi})] = \pi(\mathbf{y}^{(i)}|\mathbf{c}^{(i)}, \boldsymbol{\kappa}, \boldsymbol{\xi}), \tag{5}$$

where $\mathbb{E}_{\mathbf{u}^{(i)}}(\cdot)$ means that the expectation is taken with respect to the distribution of the $\mathbf{u}^{(i)}$.

Thanks to the (assumed) unbiasedness of the estimated likelihood, the marginal of the augmented posterior is the *exact* posterior of interest

$$\int \pi(\mathbf{c}^{(i)}, \boldsymbol{\kappa}, \boldsymbol{\eta}, \boldsymbol{\xi}, \mathbf{u}^{(i)}|\mathbf{y}^{(i)})d\mathbf{u}^{(i)} = \pi(\mathbf{c}^{(i)}, \boldsymbol{\kappa}, \boldsymbol{\eta}, \boldsymbol{\xi}|\mathbf{y}^{(i)}), \tag{6}$$

even though an estimated likelihood term has been employed inside Eq 4. An efficient way to obtain an (non-negative) unbiased likelihood estimate for state-space models is to use a sequential Monte Carlo procedure known as the particle filter [60]. The particle filter approximates unbiasedly the $i$-th likelihood, that is the expectation in Eq 5 [61, 62] by using $N$ Monte Carlo draws that in this context are named "particles". The variance of this estimated likelihood decreases when increasing $N$, and typically the success of pseudo-marginal approaches relies on having a small variance for the estimated likelihood [63–65]. That is using too few particles yields inefficient inference, but on the other hand run-time increases with $N$. However, the result that makes pseudo-marginal powerful is that, from a theoretical point of view, it provides *exact* Bayesian inference [16, 59], regardless the number of particles employed, thanks to Eq 6. However, in practice, the value of $N$ impacts the efficiency of the Gibbs sampler in exploring the posterior surface, since a too small $N$ may cause the resulting Markov chain to get "stuck" into some value for many iterations (i.e. the chain becomes "sticky"). However we do not want $N$ to be too large or the algorithm may become unnecessarily expensive.

To employ as few particles as possible while encouraging the exploration of the posterior surface, the following three strategies are considered. Firstly, we induce correlation in the particles between subsequent iterations, and this still preserves exact Bayesian inference [21, 66]. However, this is only feasible for Poisson or Langevin integrators. Secondly, our framework implements a particles tuning scheme. Thirdly, guided particle proposals [9, 67] are used when possible (for details see Section A in S1 Text).

From the considerations above, PEPSDI performs exact Bayesian inference for the model parameters. However, to further reduce the computational requirements to run the inference scheme, we developed a Gibbs-sampler that targets a slightly perturbed parameterisation of a SSMEM-model and produces considerable acceleration in the runtime, and shows promising to increase scalability of the inference towards accommodating an increasing number of individuals.

## Inference for perturbed SSMEM (default option in PEPSDI)

The "perturbed SSMEM" treats cell-constant parameters $(\boldsymbol{\kappa}, \boldsymbol{\xi})$ as parameters that instead vary, with a small fixed variance, between cells. For example, for $\boldsymbol{\xi}$ this means that for the perturbed SSMEM we assume to have $\boldsymbol{\xi}^{(i)} \sim \mathcal{N}(\boldsymbol{\xi}_{pop}, \delta^2 \cdot \mathbf{I})$, with $\delta > 0$ a fixed constant selected by the

researcher (and using a similar reasoning for $\boldsymbol{\kappa}^{(i)}$). In some sense $(\boldsymbol{\kappa}, \boldsymbol{\xi})$ have been "perturbed" to artificially vary between cells. This parameterisation, inspired by how cell-constant parameters can be treated in Monolix [68], allows $(\boldsymbol{\xi}_{pop}, \boldsymbol{\kappa}_{pop})$ to be inferred alongside with the population parameters $\boldsymbol{\eta}$ in step 3 of the Gibbs sampler (Eq (2)), and $(\boldsymbol{\xi}^{(i)}, \boldsymbol{\kappa}^{(i)})$ to be inferred alongside $\mathbf{c}^{(i)}$ in step 1. Step 2 is thus avoided.

Avoiding step 2 is desirable from the computational point of view. This is because step 2 of Gibbs-sampler (Eq 2) requires a stochastic approximation of the sum of the individual log-likelihoods (for numerical stability it is preferable to work on log-transformed quantities), enabled by a particle filter. The variance of this sum is typically large, since each element of the sum is a stochastically approximated log-likelihood. To achieve a small-variance for this log-likelihood, and thus efficient inference, many particles are required which can cause substantial run-time (S3 Fig).

As seen for the Ornstein-Uhlenbeck model, the Gibbs sampler corresponding to the perturbed SSMEM occasionally produces slightly wider credibility intervals for $(\boldsymbol{\kappa}_{\text{pop}}, \boldsymbol{\xi}_{\text{pop}})$ (S6 Fig). However, we consider this a worthwhile compromise since we do not observe any bias, while reaching a speed sometimes larger than a factor 30 (S3 Fig). Moreover, if necessary PEPSDI can be run with the perturbed SSMEM to rapidly obtain preliminary results from pilot-runs, and the latter can be used to inform the setup (e.g. starting parameter values) to launch the inference for the unperturbed SSMEM.

## Testing PEPSDI

PEPSDI is written in Julia 1.5 [26] and is available on GitHub https://github.com/cvijoviclab/PEPSDI. We tested our implementation against the Ornstein-Uhlenbeck (OU) model, for which the data likelihood can be computed exactly using the Kalman filter (full details in Section D in S1 Text). Specifically, we compared both Gibbs-samplers (according to the two options in PEPSDI) against the exact inference provided by using the Kalman filter to compute the likelihood function and embedding the latter in a Gibbs sampler as in [10] (S6 Fig). We have also tested the PEPSDI stochastic solvers by comparing them with the corresponding solvers in the Julia DifferentialEquations.jl package [69].

## Single-cell microscopy data

The first set of experiments was performed with a BioPen system (Fluicell AB) as previously described [64]. In these experiments, wild-type W303–1A (MATa leu2–3/112 ura3–1 trp1–1 his3–11/15 ade2–1 can1–100 GAL SUC2) were exposed to an upshift in fructose concentration to 2.0% and 0.5% from media containing 3% ethanol and no other carbon source. The experiment was performed as following: a glass bottom Petri dish (GWST-5030, WillCo Wells) was treated with Poly-L-Lysine solution (P4832, Sigma-Aldrich) for 15 min at room temperature. The Poly-L-Lysine solution was removed, and the Petri dish was washed with MQ water two times and left to dry overnight. Yeast cells (W303(202) NRD1-mCherry- Hph MIG1-GFP-KanMX) were grown overnight to mid-log phase at 30˚C in YNB synthetic complete medium (6.7 g/l yeast nitrogen base with ammonium sulphate (formedium), 790 mg/l complete supplement mix (formedium) and supplied with 3% ethanol). These mid-log phase cells were added to the Petri dish and left to sedimen; cell which did not adhere to the surface were removed by washing with growth media. Exposure of cells to YNB media with different concentrations of fructose was performed by using a BioPen system with BioPen prime pipette tip. Imaging was performed on Leica DMi8 inverted fluorescence microscope (Leica microsystems). The microscope was equipped with a HCX PL APO 40 × /1.30 oil objective (Leica microsystems), Lumencor SOLA SE (Lumencor) led light and Leica DFC9000 GT sCMOS camera (Leica

microsystems). Three data points were taken before the cells were exposed to fructose, and after exposure, a data point was taken every 30 sec for 15 min. At every time point, five images with an axial distance of 0.5 $\mu$m were acquired in transmission and fluorescent light to ensure an in-focus image for all cells. Transmission images were acquired with 10 ms exposure and 190 LED intensity. Mig1-GFP was observed using a filtercube with an excitation: 450/490, dichroic: 495 and emission: 500–550 filtercube at 300 ms exposure. Nrd1-mCherry was observed with an excitation: 540/580, dichroic: 585 and emission: 592–668 filtercube at 160 ms exposure at 30% LED intensity. Corresponding pixel intensities of brightfield images acquired above the focal plane were divided by pixel intensities of images acquired below the focal plane using custom Matlab script. This step removes the uneven illumination and enhances diffraction pattern of the cells. Obtained cells where segmented and fluorescent signal was quantified from the GFP images with CellX software [70]. mCherry images were used as a control to mark nuclear localisation. The Mig1 localisation was calculated as (Mig1n-Mig1c)/Mig1c from the CellX output. Tracking the cells over time was performed through a custom Matlab script as described in [71].

The second set of experiments was performed on W303(202) hxk1Δhxk2Δ (YSH202, hxk1Δ::HIS3 hxk2Δ::LEU2) expressing pFRP2138 (P414GPD, TDH3p-HXK1-CYC1t TDH3p-mAmetrine BamHIlinker ADH1tail-CYC1t); strain origin and plasmid construction are described in [43]. Cells were incubated in a microfluidic chip for long-term imaging [72]. Within the microfluidics chip the cells were exposed to an upshift in fructose from 0 to 2.0%. Microfluidic setup and usage have been extensively described in [43]. Images segmentation, quantification and processing was done as described above. Mig1-GFP localisation signal was used to calculate the nuclear Mig1 index in the first 25 min. The mAmatrine signal was used to calculate the relative Hxk1 expression.

## Supporting Information

**S1 Fig. Additional results for the stochastic gene-network regulated by the circadian-clock. A**) Marginal posterior from the inference run in Fig 2 for the correlation matrix $\Phi$ (non-diagonal values of the covariance matrix $\mathbf{\Omega}$). The correlation matrix characterises the correlation between the individual parameters ($c_1$, $c_2$, $c_3$, $c_4$). The black line represents the true-value. **B**) Pair plots for the three scale parameters that were problematic to infer, ($\tau_1$, $\tau_2$, $\tau_3$), against themselves and the log-mean values $\mu_i$. **C**) Marginal posterior for the scale parameters ($\tau_1$, $\tau_2$, $\tau_3$, $\tau_4$) when simulating and doing inference for 40 (as in Fig 2) and 100 cells. Noticeably, albeit the parameter uncertainty decreases a bias still persist in ($\tau_2$, $\tau_2$, $\tau_3$) for the considered number of cells. However we notice that when using 100 cells, except for $\tau_1$ (where the difference is small), each posterior mode gets a little closer to the ground truth.
(TIFF)

**S2 Fig. Using inferred parameters to deduce mechanistic properties for the Schlögl-model. A**) Using the inferred posterior for the Schlögl-model (Fig 3), 100, 000 cells were simulated. The cells were then split into the group having a synthesis rate $c_1$ below 1274, and above 1274. For these groups the 0.2, 0.5 and 0.8 quantiles were computed. As seen from these quantiles, cells with a lower synthesis rate ($c_1$) mainly commit to the lower cell-state (e.g low gene-expression). Meanwhile, cells with a larger synthesis rate commit to, and jump between, two different states of gene-expression. **B**) Marginal posterior pair plots for the inferred parameters.
(TIFF)

**S3 Fig. Comparing run time of the PEPSDI inference options for the Schlögl-model. A**) Comparison of run-time for the non-perturbed option (blue) where ($\mathbf{\kappa}$, $\mathbf{\xi}$) are constant

between cells, and the default perturbed option (orange), where $(\kappa, \xi)$ are slightly perturbed to vary between cells. Using the same-parameters as in Fig 3, data-sets for 20, 40, 60, 80, 100 and 200 individuals were simulated. Starting from true parameter values, the number of particles was tuned according to the tuning criteria (Section B in S1 Text) and PEPSDI was run for 10, 000 iterations. Same as for the inference in the Fig 3, we simulated intrinsic noise using the Langevin approximation, while guiding our particles using the modified diffusion bridge filter and correlating the particles with $\rho = 0.999$. For all data-sets, the particle tuning procedure suggested the use of 10 particles per individual for the perturbed-model option (orange-line), while for the non-perturbed option (blue line) the procedure suggested to use (with increasing number of individuals) 20, 20, 40, 130 and 230 particles for each individual (as described in S3 all individuals have the same number of particles for the non-perturbed sampler). The left plot shows the median run-time with max-and min values (bars) computed from three independent runs. Run-time was measured as the wall-clock time on a Dell Latitude with eight cores [Intel(R) Core(TM) i5–8365U CPU @ 1.60 GHz] running on Ubuntu 20.04. To minimise noise from other computer programs the benchmark was run on a freshly rebooted laptop with no other applications open. All the runs were performed sequentially using a single core. Due to the computational burden from performing all the runs sequentially, run-time was not measured for the case of 200 individuals for the non-perturbed option. **B**) Ratio between the blue and orange line, highlighting that the default perturbed option can be faster by more than a factor 30.
(TIFF)

**S4 Fig. Modelling of Mig1-dynamics in response to fructose addition. A-C**) Posterior visual check for the 0.05% fructose data for model structure 1 and 2 using extrinsic noise-proposals A, B and (Fig 6). The credibility intervals were obtained as in Fig 2. Model 2B compared to model 2A has wider (but not biased) credibility intervals for the 0.05% fructose data, however, only model 2B accurately describes the 2% data (Fig 6F). **C**) Posterior visual check for model structure 2 and noise proposal A, where $c_1$ and $c_2$ do not vary between cells (no upstream extrinsic noise). Without upstream extrinsic noise the model fails to describe the observed cell-to-cell variability. Credibility intervals obtained as in A-B. **D**) Marginal posterior for the model-parameters not shown in Fig 6. **E**) Marginal pair posteriors for Model 2B for the the log-means and scale parameters ($\tau$) for which the individual parameters were inferred to be correlated (corr($c_1, c_2$) and corr($Mig1c_{t0}, Mig1n_{t0}$) with colours representing fructose concentration. **F)** The same marginal posteriors as in Fig 6H from multiple inference runs (colors) with different starting values. The existence of several modes for $\mu_2^{Fru}$ (middle right) shows that the model is not fully identifiable (all parameters cannot be inferred unambiguously). However, some parameter relationships are consistent between runs. Namely, the coefficient of variation is larger in low fructose for ($c_1, c_2$) (two left plots), the log-mean $\mu_2$ is larger in low fructose (middle right), and the rates ($c_1, c_2$) are strongly correlated (right plot).
(TIFF)

**S5 Fig. Hxk1 is a source of cell-to-cell variability in Mig1 localisation. A**) Mean Mig1 ratio over 15 minutes after 2% fructose addition versus the cell-varying model parameters $c_1$ for model 2B obtained by simulating 1, 320, 000 cells. Noticeably, the cell-varying $c_1$ of which hexokinase 1 (Hxk1) is a part ($c_1 \propto [Hxk1]$) explains a part of the cell-heterogeneity in Mig1 localisation. **B**) Mean Mig1 ratio over 15 minutes after 2% fructose addition versus relative mean Hxk1 expression for 132 cells obtained from single-cell time-lapse microscopy. The linear relationship is significant (p-value $7.5 \times 10^{-6}$). The mean relative Hxk1 expression, which is likely proportional against Hxk1-expression, was computed by taking the mean of the Hxk1 expression over 240 min after fructose addition. **C**) Model predicted (bars) and observed (line)

explained cellular heterogeneity ($R^2$) in Mig1-localisation by relative Hxk1-expression and $c_1$ respectively. The line is the $R^2$-value (variability explained by regression line divided by total variability) for the linear regression in c), and the bars were computed by simulating mean Mig1 ratio (0–15 minutes after fructose addition) for 132 cells 10, 000 times, and computing the $R^2$ for the Mig1n/Mig1c versus $c_1$ linear regression for each instance.
(TIFF)

**S6 Fig. Inference results for the Ornstein–Uhlenbeck model.** Inference was performed using PEPSDI with ($\kappa$, $\xi$) cell-constant and PEPSDI with ($\kappa$, $\xi$) weakly perturbed between cells (default option). These were compared against the gold-standard case, from Wiqvist et al. [10], where a Kalman-filter is embedded into the Gibbs-sampler (Alg 2 in S1 Text) for an exact evaluation of the likelihood. It can be seen that the consequence of perturbing the model is a slightly larger credibility interval for $\sigma$, however inference for the remaining parameters is remarkably similar to the non-perturbed case.
(TIFF)

**S1 Text. Detailed description of the inference framework, simulation examples and Mig1 model. A**) Detailed information about the developed framework. **B)** Guidelines for how to run our inference framework for a user provided state-space model. **C)** A brief tutorial on model construction. **D** Detailed information on circadian clock regulated gene network, Ornestein-Uhlenbeck model and Schlögl model. **E)** Detailed information about the developed Mig1 model describing the dynamics of the ratio between nuclear Mig1 and cytosolic Mig1 (Mig1n/Mig1c) upon fructose addition to carbon starved cells.
(PDF)

## Acknowledgments

We would like to thank members of the Cvijovic lab for valuable input.

## Author Contributions

**Conceptualization:** Sebastian Persson, Marija Cvijovic.

**Data curation:** Niek Welkenhuysen, Patrick Reith, Gregor W. Schmidt.

**Formal analysis:** Sebastian Persson, Sviatlana Shashkova.

**Funding acquisition:** Marija Cvijovic.

**Investigation:** Niek Welkenhuysen.

**Methodology:** Sebastian Persson, Sviatlana Shashkova, Samuel Wiqvist, Umberto Picchini.

**Project administration:** Marija Cvijovic.

**Resources:** Marija Cvijovic.

**Supervision:** Umberto Picchini, Marija Cvijovic.

**Writing – original draft:** Sebastian Persson, Sviatlana Shashkova, Umberto Picchini, Marija Cvijovic.

**Writing – review & editing:** Sebastian Persson, Sviatlana Shashkova, Umberto Picchini, Marija Cvijovic.

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
