## [Decision Letter · Decision Letter 0]

22 Dec 2021

Dear Dr. Cvijovic,

Thank you very much for submitting your manuscript "Scalable and flexible inference framework for stochastic dynamic single-cell models" for consideration at PLOS Computational Biology.

As with all papers reviewed by the journal, your manuscript was reviewed by members of the editorial board and by several independent reviewers. In light of the reviews (below this email), we would like to invite the resubmission of a significantly-revised version that takes into account the reviewers' comments.

We cannot make any decision about publication until we have seen the revised manuscript and your response to the reviewers' comments. Your revised manuscript is also likely to be sent to reviewers for further evaluation.

Sincerely,

James R. Faeder

Associate Editor

PLOS Computational Biology

Jason Haugh

Deputy Editor

PLOS Computational Biology

Reviewer's Responses to Questions

**Comments to the Authors:**

Reviewer #1: See attachment.

In addition, the authors should proofread their manuscript carefully as there are many typographical and grammatical errors.

Reviewer #2: In this article, the authors describe PEPSDI (Particles Engine for Population Stochastic DynamIcs) a modelling framework that infers model parameters from dynamic data for single-cell models that account for both intrinsic and extrinsic noise. Their framework is a Bayesian inference framework with a new inference method capable of modeling intrinsic noise with different simulators. The authors argue that this work is important due to the need to extract mechanistic information from single-cell dynamic data (e.g. single-cell microscopy) in particular to better describe and understand cell-to-cell variability. While there are existing inference methods, they suffer drawbacks (ignore noise, assume time invariant constants, other unrealistic model assumptions) that the flexibility in this framework overcome.

The authors apply their framework to simulated biological systems (circadian clock, Schlogl bistable toy model) as well as to time-lapse microscopy data of the Mig1 transcription factor to identify drivers of heterogeneity in its localization under different nutrient conditions.

As strength of the article is that the code is freely available, and perhaps more importantly for adoption, well documented with tutorial examples.

Overall, the authors are working on an important problem, namely, how can we understand intrinsic and extrinsic noise in biological systems and infer mechanistic model parameters? Their method should be adoptable and implementable for other important biological systems, outside of yeast and the toy simulated models detailed here. I have a few comments that should be addressed prior to publication:

Major Comments:

• Why were known sources of extrinsic noise not incorporated into the Mig1/SNF1 model? For example, cell cycle, cell age, and the metabolic cycle are all know to impinge on the relevant metabolic processes.

• How much data was collected for the Mig1 experiment (n=?) I may simply be overlooking the number, but it seems like a relatively small number of cells. Particularly given the ease of collecting localization data from hundreds of cells. More generally, is there a justification for the numbers of cells used? For example, 40 simulated cells were used for the circadian clock model.

• The biological example here is relatively simple, and indeed, Hxk1 could have been predicted as the driver of heterogeneity. Is there another dataset, potentially for another biological system, to which you could apply your framework? That would be more compelling (although not necessary for publication, in my opinion)

Minor Comments:

- Please clarify your justification of using the Extrande algorithm

- Some additional details are needed in the methods describing the Mig1 experiments. Please include strain construction details or source. Also please include additional details on how Mig1 nuclear localization was quantified. Was the nuclear marker used to identify the nucleus? How were cells segmented?

- I do not see where the Hxk1 expressing plasmids are detailed in the Methods/supplementary material. Was this overexpression of Hxk1?

Reviewer #3: This is a very useful piece of work. I found the statistical analysis well explained and laid out. Distinguishing between extrinsic and intrinsic noise is challenging and important and I believe that this is a timely and useful contribution.

I have two very minor points that I would like the authors to consider, that I want to mention first:

- I would have liked to see the discussion of PEPSDI moved up at the front of the results. I found the SI very useful and detailed; and I have spend a lot of time on this material. I regard this as outstanding and very clear.

- I think that the discussion of extrinsic noise could be sharpened up and there are references that could be usefully added to give a comprehensive picture of the role of extrinsic noise and how we can identify it's contribution to cell to cell variability.

Detailed comments:

pg 3, 2nd paragraph: [13] is an odd choice to dismiss alternatives to Extrande. It is too generic in my view even though it applies generally. I think Schnoerr et al (doi:10.1088/1751-8121/aa54d9) is more suitable in this context.

pg 3, 8th paragraph: how was runtime for PEPSDI determined?

Fig 2: I would like to learn more about the joint densities over inferred parameters. In my work I have found e.g. all pairwise plots useful and especially for \\tau_2-\\tau_4 I would like to see such plots (also in relation to the \\mu parameters). Similar for Figure S1.

Fig 3 (and corresponding section on pg3,4): what were the parameters chosen for the MCMC scheme chosen here? S3 contains some detail, but are 10,000 iterations enough?

pg 4-5: the examples are outlined in welcome brevity, but I would have liked a clearer discussion of where extrinsic noise enters (see e.g. the discussion in Filippi et al (doi: 10.1016/j.celrep.2016.05.024)

pg 6-7 discussion: The discussion here struck me as slightly superficial. I think the authors' contribution is not done sufficient justice. A clearer discussion of the roles of extrinsic noise could help. It is fiendishly difficult to measure extrinsic noise experimentally and distinguish it from intrinsic noise(Fu & Pachter, doi: 10.1515/sagmb-2016-0002). Dual reporter assays (Hilfinger and Paulsson 10.1073/pnas.1018832108), are difficult to implement; and while other methods are becoming available (Ham et al doi: 10.7554/eLife.69324; Gorin and Pachter doi: 10.1101/2020.09.25.312868 ) statistical inference of models such as performed here will be our best bet to get a handle on extrinsic noise.

Michael Stumpf

**Have the authors made all data and (if applicable) computational code underlying the findings in their manuscript fully available?**

Reviewer #1: Yes

Reviewer #2: Yes

Reviewer #3: Yes

PLOS authors have the option to publish the peer review history of their article (what does this mean?). If published, this will include your full peer review and any attached files.

Reviewer #1: No

Reviewer #2: No

Reviewer #3: **Yes: **Michael P.H. Stumpf
---

## [Decision Letter · Decision Letter 1]

5 Apr 2022

Dear Dr. Cvijovic,

We are pleased to inform you that your manuscript 'Scalable and flexible inference framework for stochastic dynamic single-cell models' has been provisionally accepted for publication in PLOS Computational Biology.

Best regards,

James R. Faeder

Associate Editor

PLOS Computational Biology

Jason Haugh

Deputy Editor

PLOS Computational Biology

Reviewer's Responses to Questions

**Comments to the Authors:**

Reviewer #1: The authors have done a good job of revising the manuscript to take into account the comments made by myself and the other reviewers. I would only suggest that they check the manuscript carefully as I noticed a number of grammatical/typographical errors in the revised version.

Reviewer #2: The authors have addressed my comments from my previous review. Specifically they have:

- Clarified how noise enters into the model

- Clarified justification of using the Extrande algorithm

- Included crucial experimental details (Mig1, Hxk1 plasmids)

The authors did not address my comment regarding the simplicity of the experimental system, however, I consider this nevertheless appropriate for the current publication (and state so in my previous review).

I feel that the current manuscript is appropriate for publication.

**Have the authors made all data and (if applicable) computational code underlying the findings in their manuscript fully available?**

Reviewer #1: Yes

Reviewer #2: Yes

PLOS authors have the option to publish the peer review history of their article (what does this mean?). If published, this will include your full peer review and any attached files.

Reviewer #1: No

Reviewer #2: No

---

## [Editor Report · Acceptance letter]

11 May 2022

PCOMPBIOL-D-21-01912R1 

Scalable and flexible inference framework for stochastic dynamic single-cell models

Dear Dr Cvijovic,

I am pleased to inform you that your manuscript has been formally accepted for publication in PLOS Computational Biology. Your manuscript is now with our production department and you will be notified of the publication date in due course.

With kind regards,

Livia Horvath
